# Portable Digital Monitoring System for Sarcopenia Screening and Diagnosis

**DOI:** 10.3390/geriatrics7060121

**Published:** 2022-10-25

**Authors:** Eduardo Teixeira, Lucimére Bohn, José Pedro Guimarães, Inês Marques-Aleixo

**Affiliations:** 1Research Center in Physical Activity, Health and Leisure (CIAFEL), Faculty of Sports, University of Porto (FADEUP), and Laboratory for Integrative and Translational Research in Population Health (ITR), 4200-450 Porto, Portugal; 2Faculty of Psychology, Education and Sports, Lusófona University of Porto, 4000-098 Porto, Portugal; 3Interdisciplinary Centre for Education and Development (CeiED), Lusófona University of Lisbon, 1700-284 Lisbon, Portugal; 4Escola Superior Desporto e Lazer, Instituto Politécnico de Viana do Castelo, 4960-320 Viana do Castelo, Portugal

**Keywords:** detection, quantification, strength, body composition, muscle, frailty, older adults

## Abstract

Sarcopenia is a well-known highly prevalent muscle disease that severely impairs overall physical performance in elders, inducing a massive health-related economic burden. The widespread screening, diagnosis and treatment of sarcopenia are pivotal to restrain the disease progression and constrain its societal impact. Simple-to-use, portable, and reliable methods to evaluate sarcopenia are scarce, and sarcopenia-related assessments are typically done in several time-consuming stages. This study presents a portable digital system that enables a simple and intuitive method to evaluate sarcopenia—based on the European Working Group on Sarcopenia in Older People 2 (EWGSOP2) algorithm—including the four Find-Assess-Confirm-Severity (FACS) steps. The system comprises a mobile application (app); two wireless devices: a dynamometer (Gripwise) and a skinfold caliper (Lipowise); and a back-end website. To find cases, the SARC-F questionnaire is applied. To assess sarcopenia, the handgrip strength and the sit-to-stand tests are performed with the Gripwise and an application-embedded stopwatch, respectively. To confirm cases, anthropometric measures are performed, and muscle quantity is estimated with Lipowise. Finally, to assess severity, the app stopwatch grants the gait speed test application, evaluating physical performance. This step-by-step sarcopenia assessment results in a final grading according to the cut-off points of the EWGSOP2 criteria. All data is automatically encrypted and exported into a GDPR-compliant cloud platform, in which healthcare professionals can access and monitor their patients through the internet.

## 1. Introduction

The scientific interest in sarcopenia relies mostly on its relation with adverse health outcomes [1]. In geriatric populations, sarcopenia is one of the key factors for the loss of physical independence [2] and for the increased odds for falls and premature mortality [3,4,5]. With global aging, an increase of approximately 34% in health-related costs due to the hospitalization of older adults is expected, which will be partially related to sarcopenia [6]. Thus, the widespread screening, diagnosis and treatment of sarcopenia is pivotal not only to preventing this disease but also to restraining its progression.

Although sarcopenia does not have unanimous scientific definition, diagnosis criteria or treatment guidelines [1], it has been recognized as a disease since 2016 [7]. According to current international consortiums [8,9,10,11], the identification of sarcopenia requires the presence of diminished muscle strength and muscle mass. The European Working Group on Sarcopenia in Older People 2 (EWGSOP2) encouraged the utilization of an algorithm to diagnose and to quantify sarcopenia severity [10]. This algorithm incorporates four steps: (1) find cases; (2) assess; (3) confirm; and (4) determine severity (FACS). For case finding, the EWGSOP2 recommends using the SARC-F questionnaire, a 5-item self-report questionnaire based on the elderly perception of individual limitations in strength, walking ability, rising from a chair, stair climbing and experiences with falls [12]. Then, sarcopenia assessment requires the evaluation of muscle strength, including the grip strength test [13] and the sit-to-stand test (time needed for five repetitions of sit-to-stand motion) [14]. Sarcopenia confirmation includes skeletal muscle quantity (mass) evaluation by either: (a) computed tomography or magnetic resonance imaging [15]; (b) dual-energy X-ray absorptiometry (total body lean tissue mass [16] or appendicular skeletal muscle mass (ASMM) [17]); (c) bioelectrical impedance analysis (total body lean tissue mass or ASMM estimation [18]); and (d) calf circumference [19], used as a diagnostic proxy for older adults muscle mass in settings where no other diagnostic methods are available. Finally, to determine sarcopenia severity, physical performance tests (e.g., gait speed [20]; a 4-meter typical walking speed test [21]; a short physical performance battery [15]; a timed up-and-go test [22]; or a 400-meter walk or long-distance corridor walk [23]) are carried out.

Considering the aforementioned algorithm items, sarcopenia evaluation needs to be considered at a personalized level, since the variables that characterize the disease comprise individual aspects [24]. In addition, sarcopenia-related assessments are typically done in several time-consuming stages, hampering its assessment in clinical settings and of large samples. Although there are some innovative systems being developed and tested, sarcopenia screening tools that are simple-to-use, portable, and reliable are scarce. Thus, a portable digital system enabling a simple, easy and intuitive routine to evaluate sarcopenia, following the EWGSOP2 algorithm [10], in an integrated and portable way is presented.

## 2. System Architecture and Components

Currently, several tools are available to assess sarcopenia in different clinical circumstances. Most of the diagnostic tools available to confirm sarcopenia cases consist of expensive body composition assessment methods that have highly technical requirements, such as magnetic resonance imaging (MRI) or spectroscopy (MRS), computed tomography (CT), dual-energy X-ray absorptiometry (DXA), or ultrasound, and may not be valid or accurate (e.g., bioelectrical impedance analysis (BIA) [25]), and might even be invasive, for example, a muscle biopsy [26]. Easy-to-use integrated solutions enabling a fast, non-invasive, and cost-effective screening, as well as the diagnosis and monitoring of sarcopenia are lacking. This presently available portable technological system aims to evaluate and monitor sarcopenia through the EWGSOP2 algorithm [10]. The system architecture and its components are depicted in Figure 1.

The system architecture includes four main components: (1) a mobile application for android and/or iOS devices; (2) two Bluetooth Low Energy (BLE) digital devices: a digital dynamometer (Gripwise) for handgrip strength evaluation and a digital skinfold caliper (Lipowise) for body composition assessment through skinfold thickness evaluation; (3) the back-end application with automated encryption and data processing to extrapolate significant information, which will compile and export data into a general data protection regulation (GDPR) compliant cloud platform; and (4) the website application, in which healthcare professionals can remotely access and monitor patient data, and which acts as the front-end application of the whole system, allowing to search the database and display both the patient’s personal and health data. All the collected data are then sent to two different cloud storages: the encrypted measurements data storage (Portuguese servers) and the encrypted identification data storage (German servers), to comply with the GDPR regulations.

## 3. System Workflow

As depicted in Figure 2, the system workflow was adapted to match the FACS algorithm (find cases; assess; confirm and determine severity), allowing a user-friendly experience and step-by-step sarcopenia assessment (6 steps in total) with a final grading according to the cut-off points of the EWGSOP2 criteria [10].

Initially, to find cases, i.e., in the “tracking” step (1st of 6 steps), the SARC-F questionnaire is presented and applied to patients. This 5-item self-report survey embedded in the mobile application — based on the elderly perception of their individual physical limitations — screens the risk of sarcopenia-associated adverse outcomes comprising five factors: strength, walking ability, rising from a chair, stair climbing and experiencing falls (Figure 3).

Each factor scores from 0 to 2 points, and the survey result ranges from 0 (best result) to 10 points (worst result). Strength is rated by asking how difficult it is for the person to lift or carry 10 pounds (approximately 4.5 kg), with the options of response being: 0 = no difficulty, 1 = some difficulty and 2 points = a lot of difficulty or unable to do it. Walking ability is rated by asking how difficult it is for the person to walk across the room and whether they use aids or need help to do it, with the options of response being: 0 = no difficulty, 1 = some difficulty, and 2 points = a lot of difficulty, use aids, or unable to do it without help. Rising from a chair is rated by asking how difficult it is for the person to transfer themselves from a chair or bed and whether they use aids or need help to do it, with the options of response being the same as the previous factor. Stair climbing is rated by asking how difficult it is for the person to climb a flight of 10 steps, with the options of response being: 0 = no difficulty, 1 = some difficulty, and 2 points = a lot of difficulty, or unable to do it. Lastly, experiencing falls is valued with 2 points for those who reported falling 4 times or more in the past year, 1 point for those who reported falling 1 to 3 times in the past year, and 0 points for those who did not report any falls in the past year [12]. As acknowledged by the EWGSOP2, the SARC-F questionnaire has very high specificity to predict low muscle strength but low-to-moderate sensitivity to predict sarcopenia [10]. SARC-F is also inadequate for finding sarcopenia cases when applied to patients with musculoskeletal diseases (of the knee, hip, or spine) [27], older adults at community-based healthcare activities [28], and geriatric rehabilitation inpatients [29]. Considering these concerns the EWGSOP2 also suggests the Ishii screening test as a possible case-finding tool in clinical practice [30]. Irrespective of the case-finding tool used, this system does not assume SARC-F questionnaire results as a mandatory nor an exclusionary factor when applying the FACS algorithm. The healthcare professional may (and should) proceed with the overall assessment even for SARC-F negative (having less than 4 points) results.

To assess strength, i.e., to directly evaluate if the person has “low muscle strength”, the healthcare professional must apply the handgrip strength test (cut-off points: < 27 kg for men and < 16 kg for women) in the 2nd step of the workflow (depicted in Figure 4) with the Gripwise device, or the five-repetition sit-to-stand test (cut-off points: > 15 s for men and women) using an application-embedded stopwatch (3rd step, depicted in Figure 5). The handgrip strength evaluation method follows the Southampton protocol [13]—participant seated with forearms on the armrests; wrists in a neutral position with thumbs facing upwards; vocal encouragement: “I want you to squeeze as hard as you can for as long as you can until I say stop: squeeze, squeeze, squeeze, stop”; 3 trials in each hand (alternating sides); maximal grip score used—using the Gripwise device instead of the Jamar hydraulic hand dynamometer. Gripwise—a small, light, and portable wireless device—has been developed to accurately measure handgrip strength via BLE communication, with good sensitivity, resolution and precision (for technical details, please see reference [31]), and has already been validated against the Jamar [32] (commonly considered the gold standard device for handgrip strength evaluation). The five-repetition sit-to-stand test evaluation method follows the commonly used protocol [33] in which participants sit on the edge of an armless chair with arms crossed against their chest and with hips, knees, and ankle joints at approximately 90 degrees. Participants are instructed to stand upright as quickly as possible (with a full extension of the hips and knees) and then to sit back down to the initial position. Participants are encouraged to practice the correct technique in two trials. After a one-minute rest, participants complete three repetitions (one minute of rest between repetitions) and the best score is used for analysis.

The possibility to administer the two strength tests (handgrip strength and five-repetition sit-to-stand test) is intended to facilitate and enrich the healthcare professional approach, individualizing the strength evaluation when needed. Notwithstanding, in cases where the healthcare professional administers the two strength tests, the system will always consider the worst result for the sarcopenia assessment.

To confirm cases, i.e., to assess if the person has “low muscle quantity”, anthropometric measures—calf, thigh and relaxed arm circumferences—have to be collected and inserted by the healthcare professional (4th step of the workflow, depicted in Figure 6).Next, a body composition calculation is performed (5th step of the workflow, depicted in Figure 7) with the Lipowise device, allowing an automated, simple, and quick assessment of the biceps, triceps, thigh and calf skinfold thickness. The Lipowise device is a BLE digital skinfold caliper, a validated DXA proxy [34], that allows an automated, simple, and quick assessment of body composition based on anthropometric measurement.

The EWGSOP2 recommends that skeletal muscle mass or quality be measured mainly via DXA, MRI, MRS, CT, or BIA [10]. However, all these techniques (except for BIA) require expensive and unportable equipment and highly trained clinical staff, making it difficult to adopt them systematically in clinical settings. Even more difficult is to apply them in field contexts, such as geriatric institutions, daycare centers, and community-based healthcare activities. Even though BIA is less expensive and portable, it requires that subjects fulfill some criteria before testing, such as several hours of fasting and not having a history of severe medical or mental conditions, and may overestimate skeletal muscle mass in older adults [25]. Even though simple anthropometric measures such as limb circumferences and skinfold thickness, which ideally require certified formation and training, might present a challenge in clinical and field contexts, they are simple to obtain and do not demand expensive equipment. This system uses anthropometric data to estimate total body skeletal muscle mass according to Lees’ prediction model [35]. This model was adopted by the international society for the advancement of kinanthropometry (ISAK) as the reference model to estimate skeletal muscle mass in adults. The EWGSOP2 does not indicate sarcopenia cut-off points for total skeletal muscle mass [10]. Instead, the cut-off points only refer to appendicular skeletal muscle mass [36] or total lean mass relative to height (kg/m^2^) [37]. Since the EWGSOP2 adopted these cut-off points (total lean mass relative to height) from the Geelong osteoporosis study [37], and considering that lean mass measured by DXA is highly correlated with skeletal muscle mass measured by MRI (r = 0.94 for the total body and 0.91 for the leg [38]), this system also adopted the same study values (*T* score -2.0) for the relative total lean mass (kg/m^2^) 2 standard deviations below the young adult reference mean (cut-off points: < 15 kg/m^2^ for men and < 12 kg/m^2^ for women; see Table 3 of reference [37]).

Finally, to determine severity, the stopwatch embedded in the mobile application allows the easy application of the 15 feet (approximately 4.6 m) gait speed test (6th and final step of the workflow, depicted in Figure 8), evaluating physical performance, as recommended.

This step-by-step sarcopenia assessment results in a final grading according to the cut-off points of the EWGSOP2 criteria [10]. All data will be automatically encrypted and exported into a GDPR compliant cloud platform, enabling healthcare professionals to remotely access and monitor their patients through the website application.

## 4. System Development, Limitations and Challenges

Widespread assessment of sarcopenia in diverse healthcare sectors is crucial to avoid disease aggravation and the exacerbation of sarcopenia-related comorbidities. This system—designed and developed by a team consisting of two nutritionists, two clinical nutritionists (one Ph.D.); one mechanical engineer with technical support from the Faculty of Engineering of Porto University, and an external team of four software developers—was conceived to facilitate and encourage sarcopenia assessment in different healthcare settings, such as the hospitals, nutritional clinics, pharmacies, gymnasiums, and municipal healthcare-related services. The healthcare professionals (mainly medical practitioners and nutritionists) involved in the early stages of the system’s preliminary tests were encouraged to use it in their day-to-day practices and to report the main issues of the system workflow. The system’s information communication and integration protocols include the health level seven international and the fast healthcare interoperability resources (HL7/FHIR protocol), enabling direct interaction with several European healthcare systems. Currently, the system integrates with Health Cluster Portugal (https://www.healthportugal.com/en/ (accessed on 10 September 2022)), OneCare (https://onecare.pt/en/ (accessed on 10 September 2022)) services, the European Gatekeeper (https://www.gatekeeper-project.eu/ (accessed on 10 September 2022)) project. Several solutions continuously being implemented will facilitate its integration with other healthcare systems.

Still, this system presents the following limitations. Despite being used in some ongoing pilot studies in different settings (municipalities, daycare centers, and nutrition clinics) the system has not been validated yet against the gold standard clinical methods. Considering the case-finding tool, as earlier explained, the SARC-F questionnaire is neither a mandatory nor an exclusionary factor when applying the FACS algorithm. Indeed, the proposed EWGSOP2 operational definition of sarcopenia does not include any case-finding tool. Low muscle strength is defined as the criteria for diagnosis, low muscle quantity or quality is defined as the criteria of diagnosis confirmation, and low physical performance is defined as the criteria of sarcopenias’ severity [10]. If these three factors are all met, sarcopenia is considered severe. Another limitation of the system is using anthropometric measures to confirm sarcopenia and Lees’ prediction model [35] to estimate total body skeletal muscle. This prediction model did not include older adults. Since this population is likely the most studied, the system may overestimate the confirmation of sarcopenia cases. Using cut-off points based on total lean mass [37] instead of skeletal muscle mass may also contribute to overestimating sarcopenia cases. The system will most likely also integrate calf girth analyses in future versions, since this simple anthropometric measure adequately estimates skeletal muscle mass [39,40] and poor physical performance [19]. In fact, calf girth cut-off points for sarcopenia assessment have already been proposed [41].

Some technological systems to evaluate sarcopenia have been recently developed [42,43,44,45], but these are rarely available in the market and their validity has yet to be established. Although the presented system still uses manual anthropometric evaluations (i.e., body circumferences and skinfolds), which represents an important limitation amongst older adults because of their skin and fat tissue properties [46], anthropometric evaluations do have economic and time-consuming advantages compared with current lab methods (MRI, MRS, CT, and DXA). Moreover, since healthcare professionals can remotely access and monitor their patient’s data, this system will also contribute to an increase in the adherence of sarcopenia screening and treatment management.

## 5. Summary and Conclusions

Here we present a simple, easy and intuitive portable digital system to evaluate sarcopenia, following the EWGSOP2 algorithm. This system allows the diagnosis and monitoring of sarcopenia through six simple steps. The presented system not only allows healthcare professionals to longitudinally monitor the impact of long-term sarcopenia treatments (e.g., medical, nutritional and exercise interventions) but also may contribute to increasing public knowledge about the main characteristics and risk factors for the onset and development of sarcopenia. By contributing to the widespread screening and diagnosis of sarcopenia, this system can be an important tool to prevent and attenuate the disease progression and to constrain its social and economic impact. 

## Figures and Tables

**Figure 1 geriatrics-07-00121-f001:**
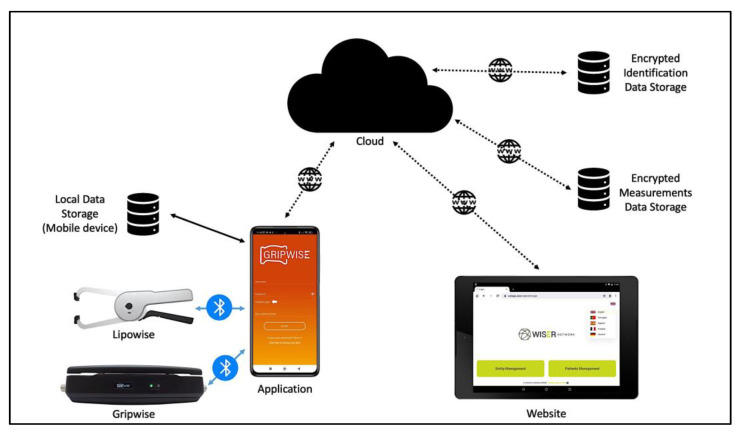
System Architecture. Lipowise and Gripwise devices (commercially available at https://wiser-net.com/shop (accessed on 10 September 2022)) communicate with mobile phone via Bluetooth Low Energy (BLE) technology. Mobile applications for iOS ((https://apps.apple.com/pt/app/gripwise/id1595974632 (accessed on 10 September 2022)) and Android (https://play.google.com/store/apps/details?id=com.gripwise.pro&hl=en&gl=US (accessed on 10 September 2022)) operating systems are currently available. Website (https://webapp.wiser-net.com/login (accessed on 10 September 2022)) is the back-end interface, i.e., the data access layer.

**Figure 2 geriatrics-07-00121-f002:**
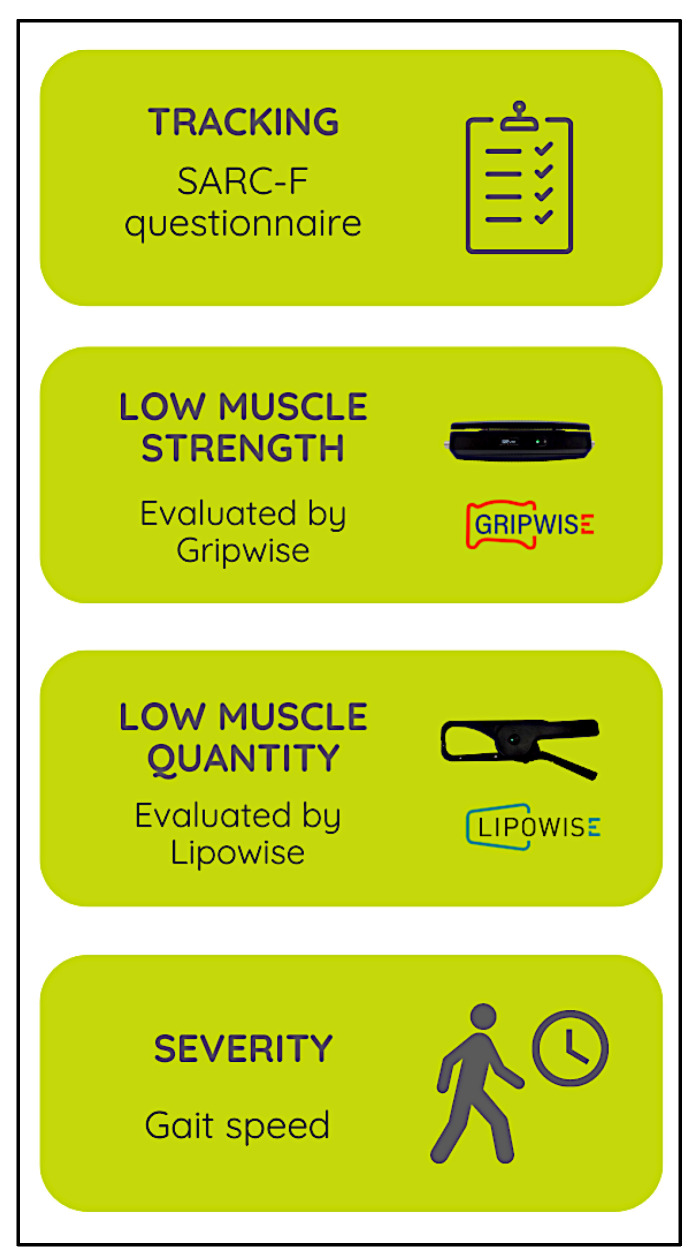
System Workflow.

**Figure 3 geriatrics-07-00121-f003:**
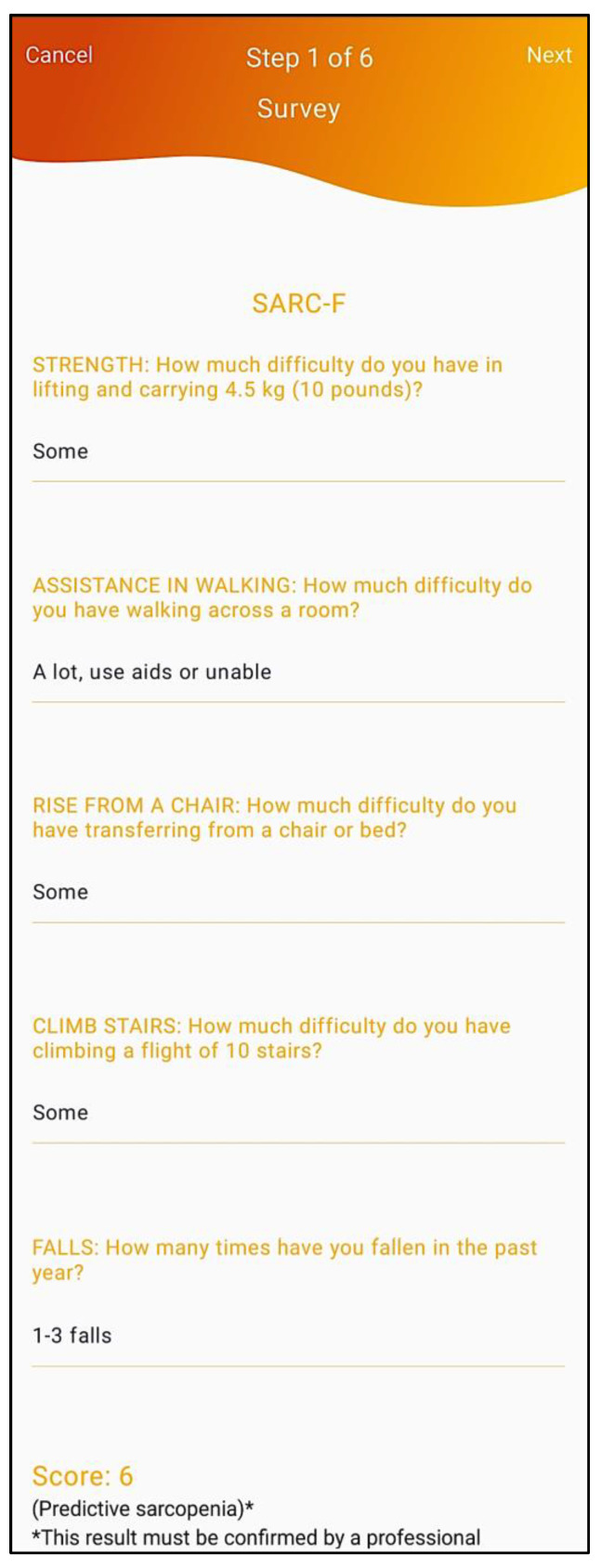
Application of the SARC-F questionnaire.

**Figure 4 geriatrics-07-00121-f004:**
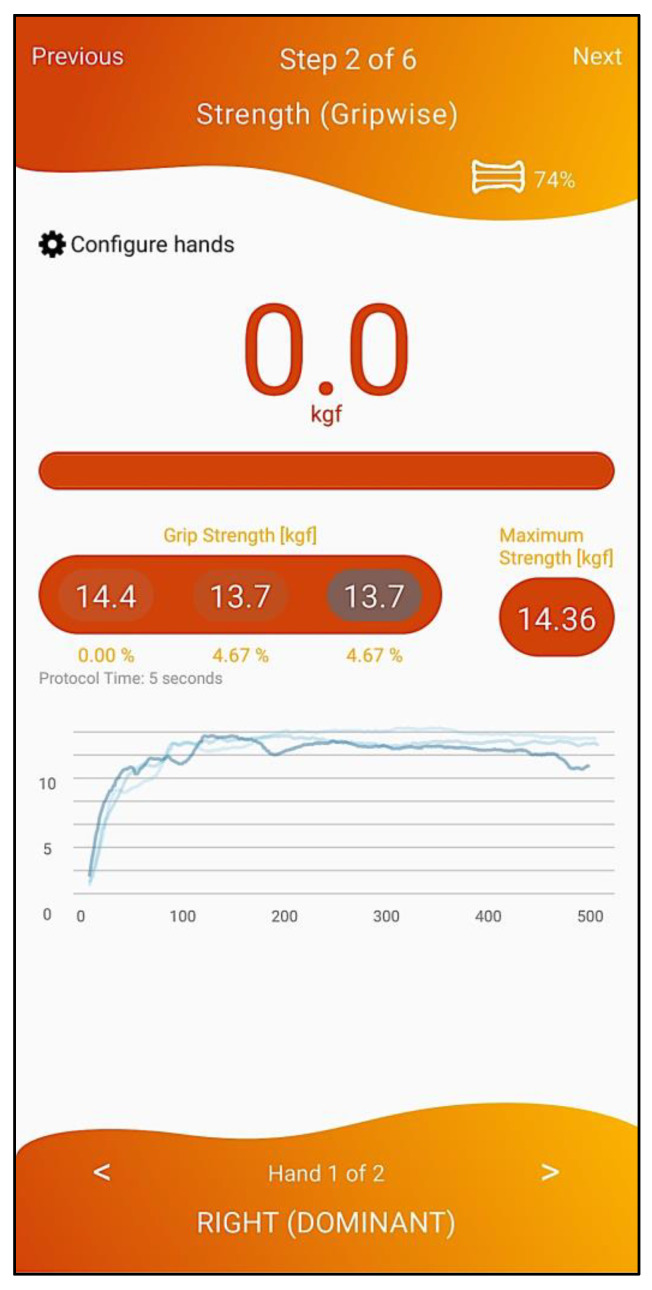
Handgrip strength evaluation with Gripwise.

**Figure 5 geriatrics-07-00121-f005:**
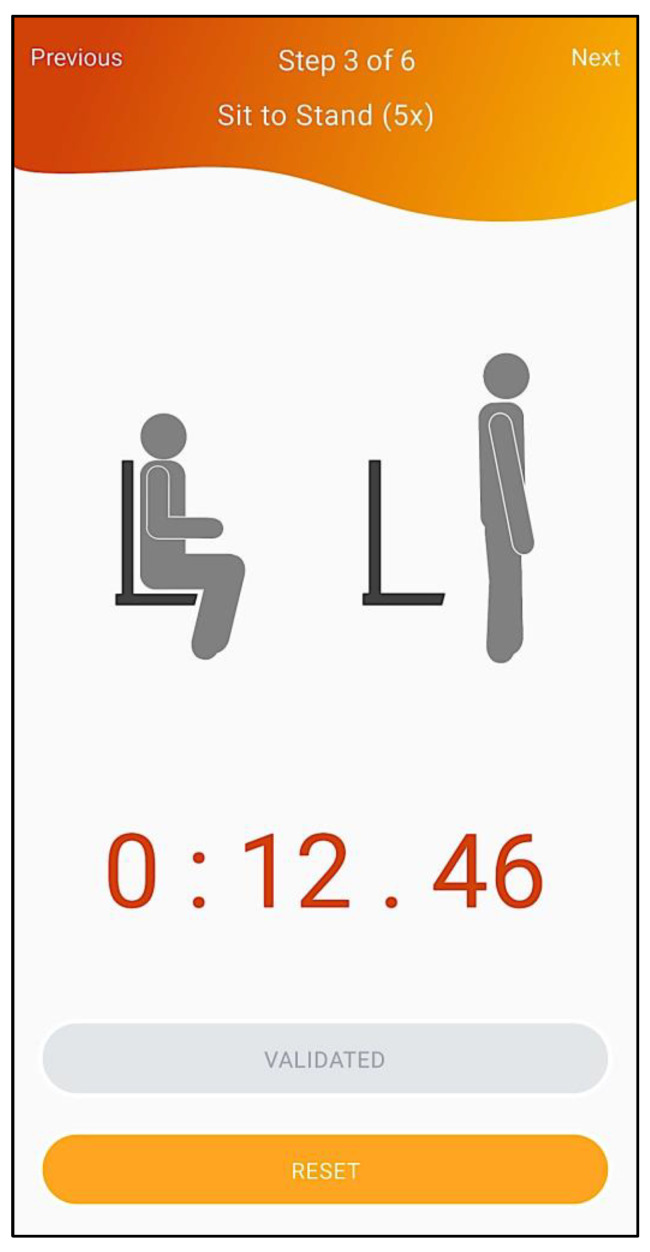
Five-repetition sit-to-stand test.

**Figure 6 geriatrics-07-00121-f006:**
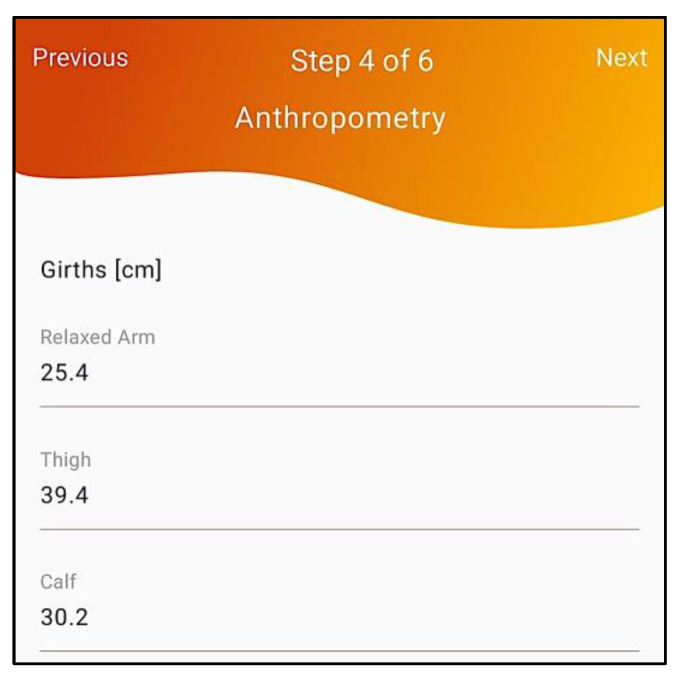
Anthropometric measures evaluation.

**Figure 7 geriatrics-07-00121-f007:**
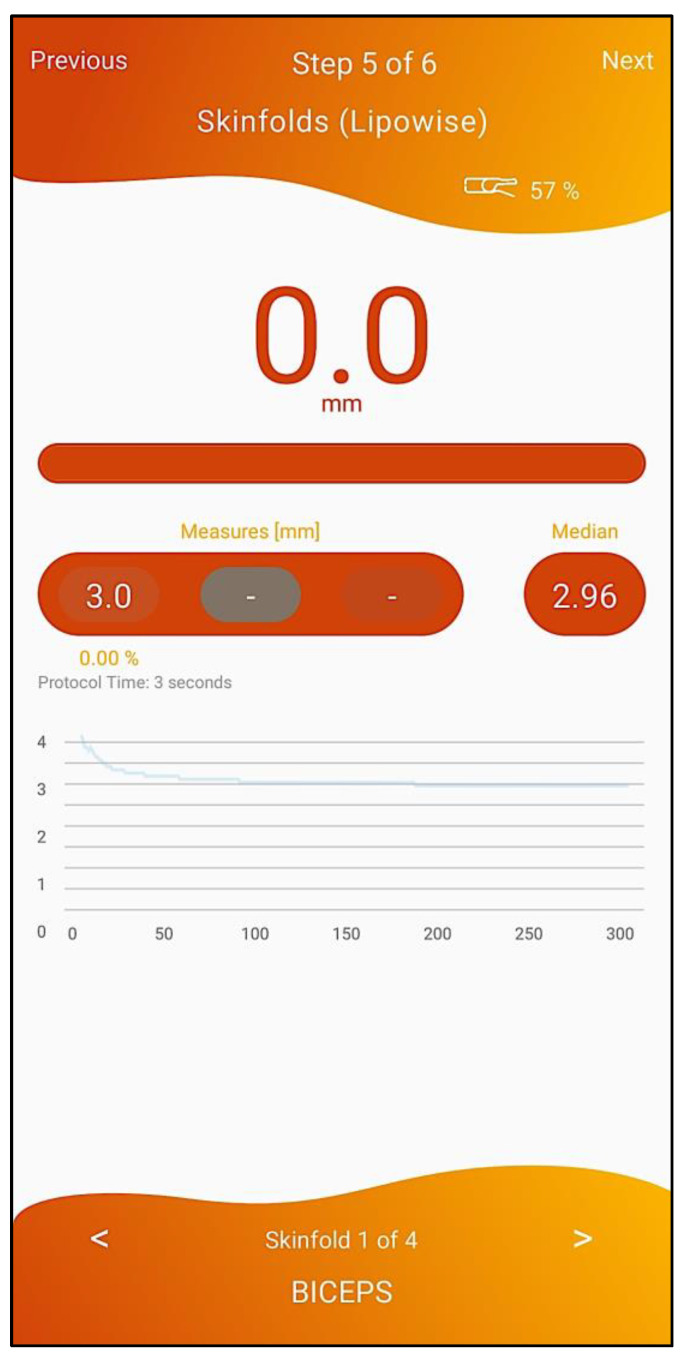
Biceps skinfold thickness evaluation with Lipowise.

**Figure 8 geriatrics-07-00121-f008:**
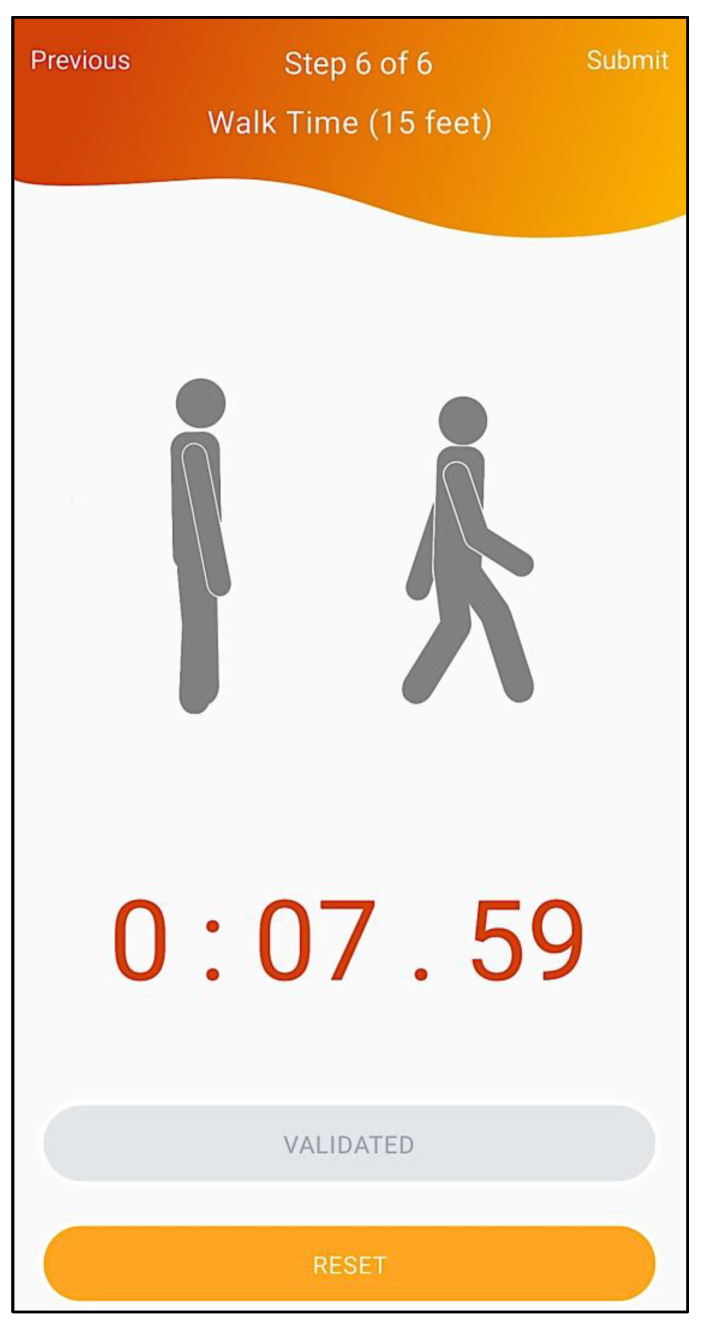
Gait speed test.

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
