# Peer review of "Portable Digital Monitoring System for Sarcopenia Screening and Diagnosis"

_geriatrics, 2022, doi:10.3390/geriatrics7060121_

Round 1

Reviewer 1 Report

This study introduced a portable digital system for an easy and fast evaluation of sarcopenic phenotypes in aging human skeletal muscles. The manuscript is largely well-written, however not in a rigorous scientific format. In regards to the evaluation of sarcopenia by this systems, I have a few major concerns:

1.       In the SARC-F questionnaire, as mentioned in line 107-110, there are only three points, 0, 1, 2, for each question. My concern would be that the sarcopenia is a progressive phenotype in aging, and most of the times people might encounter the situation more than "some difficulties", but less than "a lot of difficulties or unable to do so", so I would assume if more scales are provided in the options, it would be more specific to evaluate the precise situation of the phenotype.

2.       Sarcopenia as a chronic progressive phenotype, it can occur at a relatively early age point, i.e. 45-50, and progresses for decades to cause the actual morbidity or mortality. Based on that, the questions set in the SARC-F questionnaire are only suitable for the middle or later stage of sarcopenia. Does it have the capability to detect any early onset of sarcopenia? For example, the patient might not have difficulties of doing such action, but the muscle strength has been significantly lowered than healthy subject.

3.       As a diagnostic system, have the authors ever tested this system with actual subject? Any actual data to show the diagnosed sarcopenia using this system, any difference in the severity of the phenotype?

Author Response

Dear Reviewer,

The authors thank all the pertinent comments and concerns raised about our manuscript. A point-by-point response follows:

- This study introduced a portable digital system for an easy and fast evaluation of sarcopenic phenotypes in aging human skeletal muscles. The manuscript is largely well-written, however not in a rigorous scientific format. In regards to the evaluation of sarcopenia by this systems, I have a few major concerns:

Point 1: In the SARC-F questionnaire, as mentioned in line 107-110, there are only three points, 0, 1, 2, for each question. My concern would be that the sarcopenia is a progressive phenotype in aging, and most of the times people might encounter the situation more than "some difficulties", but less than "a lot of difficulties or unable to do so", so I would assume if more scales are provided in the options, it would be more specific to evaluate the precise situation of the phenotype.

Response 1: We agree with the reviewer's comment and acknowledge that the SARC-F questionnaire may present some limitations. These limitations are also acknowledged by the EWGSOP2. Still, this manuscript does not aim to modify and critically evaluate the SARC-F questionnaire. Despite its caveats, the SARC-F questionnaire was adopted by de EWGSOP2 as the tool to find sarcopenia cases. This system only adopted their recommendations. A paragraph addressing the SARC-F questionnaire issues was added to chapter 3 (lines 169-178). A discussion about this topic was also added in chapter 4 (lines 439-445).

Point 2: Sarcopenia as a chronic progressive phenotype, it can occur at a relatively early age point, i.e. 45-50, and progresses for decades to cause the actual morbidity or mortality. Based on that, the questions set in the SARC-F questionnaire are only suitable for the middle or later stage of sarcopenia. Does it have the capability to detect any early onset of sarcopenia? For example, the patient might not have difficulties of doing such action, but the muscle strength has been significantly lowered than healthy subject.

Response 2: Thank you for your pertinent comment. A paragraph addressing the SARC-F questionnaire issues was added to chapter 3 (lines 169-178). A discussion about this topic was also added in chapter 4 (lines 439-445).

Point 3: As a diagnostic system, have the authors ever tested this system with actual subject? Any actual data to show the diagnosed sarcopenia using this system, any difference in the severity of the phenotype?

Response 3: Thank you for the comment. We don´t have yet validated this system against gold-standard clinical measures. Nonetheless, the application has been tested several times and, despite minor bugs, it is working adequately. We have included this information in the second paragraph of chapter 4 (lines 256-258).

Reviewer 2 Report

General comments

Please clarify throughout the manuscript whether this app is intended for use by patients at home or by healthcare professionals in the clinic. Currently, it is unclear whether the tests are intended to be self-assessed or assessed by healthcare professionals.

Please provide an overview describing the steps behind how the app was designed and developed, perhaps under a new subheading. Importantly, note which stakeholders were involved in its development, including whether patients and members of the public were involved.

Please comment whether this app is commercially available (i.e. downloadable for common app stores).

Specific comments

Lines 72-73: Please give examples of the tools that are currently available to assess sarcopenia in clinical settings, and how this app overcomes their limitations.

Figure 1: Please label the digital dynamometer, skinfold calliper, and the website application in Figure 1. Currently, it is not clear what these are from only looking at the Figure and Figure caption.

Lines 99-104: Please cite and discuss studies that have validated the SARC-F questionnaire as a screening tool for sarcopenia. 

Lines 123-130: Please provide information on the criteria for “low muscle strength” based on the handgrip strength test and 5-time chair stand.

Lines 123-130: Please also provide information on how participants would perform the handgrip and chair-stand test. Are these tests intended to be self-assessed or assessed by healthcare professionals in the clinic (see my comments)? For the chair-stand test, I recommend referring to/referencing previously published sit-to-stand protocols in older adults (e.g. https://doi.org/10.1016/j.gaitpost.2019.12.003).

Can you please justify why two tests are used for muscle strength (chair stand, handgrip) and anthropometry (skinfold, circumferences). The app is intended be quick and practical – would it not be simpler to only include one test in each? And what happens if one test indicates low muscle strength and the other test indicates adequate muscle strength – which test result do you interpret and use?

Lines 135-142: EWGSOP2 do not recommend the use of skinfolds to assess body composition/muscle quality. Please acknowledge this and provide a thorough, evidence-based rationale to justify deviating away from EWGSOP2 recommendations.

Line 172: Change “diagnose” to “diagnosis”

Lines170-179 (discussion section): Please provide an overview of what the next intended steps are to validate the app and subsequently integrate it within healthcare systems. How are you going to go about this?

Author Response

Dear Reviewer,

The authors thank all the pertinent comments and concerns raised about our manuscript. A point-by-point response follows:

- General comments

Point1: Please clarify throughout the manuscript whether this app is intended for use by patients at home or by healthcare professionals in the clinic. Currently, it is unclear whether the tests are intended to be self-assessed or assessed by healthcare professionals.

Response 1: Thank you for your comment. This system is intended for the use of healthcare professionals in diverse settings, e.g., hospitals, pharmacies, nutrition clinics, gymnasiums, municipalities, etc. We included this information in the first paragraph of chapter 4 and clarified it throughout the manuscript. Chapter 4 title was updated to System Development, Limitations and Challenges.

Point 2: Please provide an overview describing the steps behind how the app was designed and developed, perhaps under a new subheading. Importantly, note which stakeholders were involved in its development, including whether patients and members of the public were involved.

Response 2: Thank you. We included this information in the first paragraph of chapter 4.

Point 3: Please comment whether this app is commercially available (i.e. downloadable for common app stores).

Response 3: Thank you. Yes, the application is currently available for iOS (https://apps.apple.com/pt/app/gripwise/id1595974632) and Android (https://play.google.com/store/apps/details?id=com.gripwise.pro&hl=en&gl=US) operating systems. This information was added to the figure 1 caption.

- Specific comments

Point 4: Lines 72-73: Please give examples of the tools that are currently available to assess sarcopenia in clinical settings, and how this app overcomes their limitations.

Response 4: Thank you for your suggestion. We inserted examples of the most common methods available in (some) clinical settings in chapter 2 (lines 73-77). The application advantages compared to these methods were included in the third paragraph of chapter 4 (lines 458-463).

Point 5: Figure 1: Please label the digital dynamometer, skinfold calliper, and the website application in Figure 1. Currently, it is not clear what these are from only looking at the Figure and Figure caption.

Response 5: Thank you for your comment. Both figure 1 and its caption have been revised and improved.

Point 6: Lines 99-104: Please cite and discuss studies that have validated the SARC-F questionnaire as a screening tool for sarcopenia.

Response 6: Thank you for your suggestion. A paragraph addressing the SARC-F questionnaire issues was added to chapter 3 (lines 169-178). A discussion about this topic was also added in chapter 4 (lines 439-445).

Point 7: Lines 123-130: Please provide information on the criteria for “low muscle strength” based on the handgrip strength test and 5-time chair stand.

Lines 123-130: Please also provide information on how participants would perform the handgrip and chair-stand test. Are these tests intended to be self-assessed or assessed by healthcare professionals in the clinic (see my comments)? For the chair-stand test, I recommend referring to/referencing previously published sit-to-stand protocols in older adults (e.g. https://doi.org/10.1016/j.gaitpost.2019.12.003).

Response 7: Thank you. The cut-off points and the measurement protocols for the handgrip strength and the 5-times chair stand tests were included (lines 179-200). The tests are intended to be performed by the healthcare professional, as stated in line 180 and revised throughout the manuscript.

Point 8: Can you please justify why two tests are used for muscle strength (chair stand, handgrip) and anthropometry (skinfold, circumferences). The app is intended be quick and practical – would it not be simpler to only include one test in each? And what happens if one test indicates low muscle strength and the other test indicates adequate muscle strength – which test result do you interpret and use?

Response 8: Thank you for your comment. The possibility to perform the two strength tests (handgrip strength and 5-times chair stand test) intends to facilitate and enrich the healthcare professional approach, individualizing the strength evaluation when needed. Notwithstanding, in cases where the healthcare professional performs the two strength tests, the system will always consider the worst result for the sarcopenia assessment. This information was added for the strength tests (lines 173-177) and for anthropometry (lines 299-322).

Point 9: Lines 135-142: EWGSOP2 do not recommend the use of skinfolds to assess body composition/muscle quality. Please acknowledge this and provide a thorough, evidence-based rationale to justify deviating away from EWGSOP2 recommendations.

Response 9: Thank you for your comment. The rationale for this topic was added in chapter 3 (lines 299-322). A discussion about this limitation was also added in chapter 4 (lines 446-454).

Point 10: Line 172: Change “diagnose” to “diagnosis”

Response 10: Thank you. Done.

Point 11: Lines170-179 (discussion section): Please provide an overview of what the next intended steps are to validate the app and subsequently integrate it within healthcare systems. How are you going to go about this?

Response 11: We intend to validate the app against gold-standard methods in a clinical setting. We have already implemented some protocols with European hospitals and primary care centers. And we also have some protocols running with municipalities and daycare centers. A paragraph was added in chapter 4 (lines 256-259) addressing this issue.

Round 2

Reviewer 1 Report

I acknowledge the revisions made by the authors regarding my first comments. I have no more concerns with these revisions.